# Malnutrition and Biomarkers: A Journey through Extracellular Vesicles

**DOI:** 10.3390/nu14051002

**Published:** 2022-02-27

**Authors:** Herminia Mendivil-Alvarado, Leopoldo Alberto Sosa-León, Elizabeth Carvajal-Millan, Humberto Astiazaran-Garcia

**Affiliations:** 1Department of Nutrition, Research Center for Food and Development, CIAD, A.C., Hermosillo 83304, Mexico; herminia.mendivilal@gmail.com; 2Independent Researcher, Hermosillo 83304, Mexico; leo.sosa@westnet.com.au; 3Biopolymers, Research Center for Food and Development, CIAD, A.C., Hermosillo 83304, Mexico; ecarvajal@ciad.mx; 4Department of Chemical and Biological Sciences, University of Sonora, Hermosillo 83000, Mexico

**Keywords:** exosomes, microvesicles, obesity, undernutrition, microparticles

## Abstract

Extracellular vesicles (EVs) have been identified as active components in cellular communication, which are easily altered both morphologically and chemically by the cellular environment and metabolic state of the body. Due to this sensitivity to the conditions of the cellular microenvironment, EVs have been found to be associated with disease conditions, including those associated with obesity and undernutrition. The sensitivity that EVs show to changes in the cellular microenvironment could be a reflection of early cellular alterations related to conditions of malnutrition, which could eventually be used in the routine monitoring and control of diseases or complications associated with it. However, little is known about the influence of malnutrition alone; that is, without the influence of additional diseases on the heterogeneity and specific content of EVs. To date, studies in “apparently healthy” obese patients show that there are changes in the size, quantity, and content of EVs, as well as correlations with some metabolic parameters (glucose, insulin, and serum lipids) in comparison with non-obese individuals. In light of these changes, a direct participation of EVs in the development of metabolic and cardiovascular complications in obese subjects is thought to exist. However, the mechanisms through which this process might occur are not yet fully understood. The evidence on EVs in conditions of undernutrition is limited, but it suggests that EVs play a role in the maintenance of homeostasis and muscle repair. A better understanding of how EVs participate in or promote cellular signaling in malnutrition conditions could help in the development of new strategies to treat them and their comorbidities.

## 1. Introduction

The term malnutrition encompasses disorders associated with deficit or excess in the consumption of nutrients, which manifests in conditions such as undernutrition and excess weight (obesity), known as the double burden of malnutrition. The figures for malnutrition worldwide are alarming, and show that it affects children and adults alike. According to the World Health Organization, in 2014, there were approximately 462 million underweight adults. It has also been reported that global obesity prevalence has risen approximately 2 percentage points per decade since 1975 [1]. Worldwide, in 2016, 678 million adults were reported to have obesity [2], and it is proposed that this amount will increase to 1.12 billion obese individuals by 2030 [3].

Obesity is defined by the World Health Organization as excess body fat, using a body mass index (BMI) greater than or equal to 30 kg/m^2^ as a reference. The double burden of malnutrition causes health issues such as muscle wasting; the propensity to develop cardiovascular, chronic-degenerative diseases; and an increase in the incidence of infections, among other maladies, which affect the health status and quality of life of the sufferers.

Currently, anthropometric tools and indicators, biochemical parameters, and biomarkers are used to assess the nutritional status of the population, aiming at the timely detection of malnutrition in some of its forms [4]. Among these tools, the use of biomarkers of nutritional status stands out. Biomarkers are measurable parameters or molecules, that can be used to assess the stability or, alternatively, the degree of abnormality of a particular biological process, making it possible to detect or monitor the deterioration of health and, in some cases, nutritional alterations. However, the predictive power of traditional biomarkers used in the field of nutrition (e.g., plasma metabolites and/or body parameters) does not adequately reflect the nutritional status of the individual [5]. They are generally late in showing results of clinical value, because their quantification varies depending on the presence of disease, disease stage, pathological condition, or metabolic alteration [6]. Thus, the possibility of early detection of malnutrition conditions or complications associated with them is limited, which in turn increases the morbidity and mortality of sufferers. Given this, it is essential to go beyond specific metabolites and to explore new mechanisms of signaling and/or cellular communication [7].

In the last decade, the study of extracellular vesicles (EVs) has gained considerable interest due to the active role they play in cellular communication. It is known that EVs are particles released by cells into the extracellular space, and that both their release and content respond to the conditions surrounding the cellular microenvironment [8]. This response seems to be a highly calibrated one, such that it promises predictability of response in the event of alterations within the microenvironment surrounding the cells. This makes the study of EVs not only fascinating, but also relevant in that, precisely because of this predictability, it allows for the possibility for detectability and measurability, thereby helping to close the gap between the deterioration of health and time to detection. Henceforth, when we speak of the sensitivity of EVs, we are referring to this seemingly highly calibrated response, which is manifested with variations in the size, quantity, and content of EVs. Indeed, due to the response of EVs to situations of cellular stress, their use as potential biomarkers in some pathological conditions has been suggested, such as in neurodegenerative, cardiovascular, and chronic-degenerative diseases and some types of cancer [9,10,11,12].

Although the literature does show that there is an association between EVs and diseases or health complications associated with obesity and undernutrition, little is known about the influence of malnutrition alone (that is, without the influence of additional illnesses) on the heterogeneity and specific content of EVs. A better understanding of how EVs participate in or promote cellular signaling in situations of malnutrition could help in the development of new strategies to treat them and their comorbidities. With this in mind, the aim of this review is to examine the current research on the effect of malnutrition on EVs and the likely role of EVs in the development of comorbidities associated with undernutrition and obesity.

This review is divided into five sections. The first section deals with the basic general knowledge about EVs and their classification. The subsequent four sections examine the current evidence on the sensitivity of EVs to specific conditions of nutritional status: excess weight, adipose tissue, diet, weight loss, and undernutrition.

## 2. Definition and Biogenesis of Extracellular Vesicles

The study of EVs is broad and branches out into many areas such as renal diseases, cancer, and autoimmune and cardiac diseases. For a better understanding of this rapidly evolving and developing field of study, it is worth reviewing the theoretical bases about its biogenesis and classification that have so far been proposed, bearing in mind that little is known about the influence of nutritional status on the sensitivity of EVs.

It is well known that cells use different signals and vehicles to transmit information to other cells [13,14]. They expel cytoplasmic and membrane material through vesicles that, when found in the extracellular medium, are called EVs. The process of the production of EVs is carried out by most cells in the body and is a phenomenon that has been maintained throughout evolution, both in eukaryotic and prokaryotic organisms [15]. The International Society for Extracellular Vesicles (ISEV) proposed, as a generic concept, the term “extracellular vesicles” for all those particles delimited by a lipid bilayer, which are naturally released from the cell to the extracellular medium, with the characteristic that they cannot replicate nor have a nucleus [14].

At first, EVs were thought of as waste carriers only. It was not until 1987 that the first hypotheses about their existence as active components in cellular communication were raised [16]. Since then, research on EVs has increased, providing information about their components, possible biogenesis mechanism, and even their classification. More recently, a debate as to the most accurate classification of EVs subpopulations took place. As part of this debate, considerations regarding their content and morphometric characteristics were at issue, as questions were raised as to whether the mechanism of biogenesis and content of EVs were dependent on the type of cell that produced them. Faced with this problem, the ISEV proposed a series of guidelines regarding nomenclature, as well as the minimum requirements to define populations of EVs [14].

The classification of EVs is based on their biogenesis and size, broadly separates them into two groups, exosomes (50–150 nm) and microvesicles (<1000 nm), also known as small and medium EVs, respectively. However, the ISEV has encouraged caution when using the terms exosomes and microvesicles, as they could be confused with terms that were historically used to refer to something else, and might be contradictory and inaccurate when referring to concepts about EVs. Such is the case for small EVs, also known as exosomes, which should not be confused with the exosomal complex [14].

### 2.1. Exosomes

Exosomes represent a group of small vesicles with sizes ranging from 50 to 150 nm [14]. They can come from any type of cell, although initial investigations have suggested that they came from hematopoietic and dendritic cells [17,18,19].

Exosomes are initially generated within the lumen of endosomes as intraluminal vesicles (ILVs), and during their maturation they undergo a process to become late endosomes, also known as multivesicular bodies (MVBs). MVBs fuse their contents with the plasma membrane of the cell and are then expelled into the extracellular medium, by the cell, as EVs [16]. Some of the mechanisms through which this process takes place are still under study. However, within the main mechanisms involved, the endosomal transport sorting complex (ESCRT—endosomal sorting complexes required for transport) and its subcomplexes (ESCRT-0, -I, -II, and -III) are known to play a role in the sorting and conformation of multivesicular endosomes (an earlier form of MVBs), as well as in the secretion and excretion of exosomes [20,21,22]. Thus, the ESCRT complex, together with its four subcomplexes, represent an important step in the formation of exosomes. It has been shown in dendritic cells that the depletion in the formation of exosomes directly affects their production [22]. This mechanism does not seem to be the only one, however. The lipid microdomains in the plasma membrane, the activity of sphingomyelinase (nSMase) [23], the presence of flotiline [24], and the affinity of proteins associated with tetraspanins [25] have been shown to also be involved in the genesis of ILVs.

The presence and enrichment of proteins in the exosomes is varied and can include proteins involved in the process of biogenesis and/or vesicular traffic. These include the family of tetraspanins (CD81, CD63, CD82, and CD9), a group of transmembrane proteins that form complexes with each other, as well as with different transmembrane and cytosolic proteins [25,26,27,28]; associated proteins such as integrins and immunoglobulins; cytoskeleton proteins (tubulin and actin); ESCRT complex-related proteins (ALIX and TSG-101) [29,30]; and heat shock chaperone proteins (HSP70 and HSP90), which are found in most exosomes [31].

Exosome biogenesis is a complex process whose understanding is made more difficult by the fact that the regulation of each of the mechanisms hitherto described remains unknown; furthermore, the possibility that they could coexist in a given cell type also cannot be ruled out completely. Likewise, the targeting of these small vesicles is not fully understood, but to date, it is known that this could depend on their content, type of originating cell, mechanism of biogenesis, and/or the pathological situation of the cell [21]. Much work remains to be done to unravel the mechanisms of biogenesis and that of the targeting of exosomes.

### 2.2. Microvesicles

Microvesicles (MVs), also known as microparticles or ectosomes, are vesicles that measure from 100 to 1000nm [14]. They were originally considered tiny particles from platelets called “platelet dust/debris”, found in the plasma and serum [32]. Today, it is known that they come from different types of cells and are vesicles generated by direct sprouting of the plasma membrane, whose process involves the reorganization of actin and the subsequent detachment of the vesicle towards the cell exterior [14]. Although the biogenesis of MVs, as it relates to their release into the extracellular medium, is yet to be fully described, it is known that different mechanisms are required to integrate the rearrangement of lipids and membrane proteins to complete this process, including calcium-dependent and independent mechanisms [33].

The content of MVs can vary. However, within their components, lipids and proteins involved in their biogenesis can be found. An example of these is the group of RHO proteins (GTPases) and RHO (rock)-associated protein kinases. As for lipids, the most enriched in the MVs are different types of lipids/phospholipids, among them lysophosphatidylcholine, sphingolipids, ceramides, and cholesterol. The components of MVs, as well as their mechanisms of genesis and traffic, are still areas under study.

In general, the composition and specific markers of MVs and exosomes are different and depend on the biogenesis of each subpopulation and the type of cell from which they come. The use of membrane markers is one of the most used techniques to classify both MVs and exosomes. However, there are proteins or markers that are shared by both groups, which makes exact identification more difficult. In an attempt to standardize the classification and composition of specific markers, the ISEV has suggested minimum information for studies of EVs, considering the use of operational terms for EV subtypes to be referenced, including their physical characteristics (size and/or density) and chemical composition; a list of EV specific markers is also suggested [14]. Despite ISEV efforts, the characterization of these subpopulations remains a challenge.

## 3. Extracellular Vesicles and Excess Weight

Obesity, an excessive accumulation of body fat (BMI ≥ 30 kg/m^2^), adversely affects body function and favors the development of comorbidities, such as cardiovascular and metabolic diseases [34,35]. The early identification of these conditions is key to timely treatment. However, there are people with obesity who do not develop metabolic disorders and show apparent health, known as “metabolically healthy obese” (MHO) [36]. The situation of the MHO, however, does not preclude future deterioration of health. In fact, it is the MHO who present a higher cardiovascular risk, as alterations in communication at the cellular level continue to occur, even without the apparent changes traditional metabolic biomarkers [37,38,39,40,41]. Due to the sensitivity of EVs to situations of metabolic stress, recent studies conducted in animals and just a few in humans on the characteristics of EVs and excess weight have shown significant and interesting advances (Table 1). Even so, research on the use of EVs as potential biomarkers in situations of obesity remains limited.

Most of the studies on EVs have been conducted in murine models without metabolic complications; only a few have been carried out in obese subjects. In both, changes in the general characteristics of EVs, size, number, and content, e.g., nucleic acids (mRNA, miRNA, etc.), have been reported. Once these changes occur, they might be the subject of further changes, depending on the degree of obesity. Furthermore, some of these characteristics and the content of EVs have been positively correlated with indicators such as BMI and biomarkers such as glucose, insulin, and serum lipids [45,49,54,55,56,57], among others. This correlation has led some authors to suggest that the characteristics of EVs (level and size) are affected by the inflammatory microenvironment caused by obesity.

Studies in obese adults have found that the characteristics of EVs could depend on the level of development of obesity, but not on its associated metabolic complications [45]. Goichot, for example, reported that the increase in plasmatic EVs concentration in obese subjects could explain their higher risk of thrombotic complications [42]. Based on this, it has been suggested that endothelial and platelet-derived EVs could be involved in the pathogenesis of endothelial dysfunction in obesity. Additionally, the increase in the number and concentration of EVs has been correlated with the increase in the insulin resistance index (HOMA-IR) [43,49,52] and associated components in the insulin signaling pathway [53], as well as with high levels of triglycerides in blood and excess body fat [44,51]. Taken together, this research suggests that EVs could be involved in the development of metabolic complications, but more evidence is needed to reveal the cellular pathway by which these changes occur, which translate into metabolic alterations.

Despite the fact that most of the evidence to date suggests that EVs increase in number and size in many body fluids in obese people, there is one study that suggests otherwise. Santamaria et al. reported that miRNA cargo of plasma EVs are associated with obesity, as well as smaller sizes of EVs in obese women than in those with normal weight [52]. However, the number of small EVs isolated from obese and lean participants was found to be equivalent in obese and normal weight women [52]. It is not clear what explains these results, but Santamaria suggested that the significant differences in glucose parameters and increased fatness in obese women were responsible for the plasma EVs changes. In agreement with Santamaria et al., Durcin suggested that the production and expulsion of EVs occurs following exposure to different biological stimuli related to the chronic low-grade inflammation state associated with obesity [54].

So, the relationship that seems to exist between the characteristics of EVs and metabolic biomarkers suggests that EVs are also clear indicators of the development of metabolic disorders or diseases. This is promising for the study of EVs as biomarkers in excess weight morbidities. However, more research is needed in order to describe the specific role that EVs play in these pathological processes, as well as their subsequent validation as potential biomarkers in humans.

## 4. Extracellular Vesicles and Adipose Tissue

The study of EVs as potential biomarkers of alterations to nutritional status includes EVs from specific cells or tissues, such as endothelial, platelet, and adipose tissue [58,59].

It appears that alterations in the adipose tissue of MHO subjects have an influence on the characteristics of extracellular platelet and endothelial vesicles, which, in turn, seem to have an effect on the development of cardiovascular and metabolic diseases [45,47]. The reason for this, as has been explained, is that the adipose tissue of adults MHO secretes cytokines alter endothelial function and this, in turn, promotes the activation of the transcription factor NF-KB [60] and pro-inflammatory pathways. This mechanism has also been shown in murine models with hypertension [61,62,63,64]. Furthermore, it has been shown that the activation of NF-KB can also be stimulated by the EVs of macrophages [65] and adipocytes [50], which induce abnormalities in the glucose−insulin balance. The insulin-dependent decrease in glucose assimilation has been reported to be due to the inhibition, at least in part, of Akt phosphorylation [46,48], which in turn interferes with the translocation of GLUT-4 in adipocytes. Given this, it has been proposed that the decrease in insulin-stimulated glucose absorption is mediated by the activation of NF-KB induced by EVs [50]. However, this is just a small part of the role that EVs play in the main metabolic pathways involved in the development of comorbidities of obesity, such as insulin resistance.

Studies using adipose tissue explants (from obese adults) have shown changes in the expression of key proteins in signaling pathways, such as TFG-B, which is involved in the development of fibrosis in various processes of chronic inflammation, especially in the liver [48]. Koeck et al. reported that EVs from adipose tissue could play an important role in the pathogenesis and development of nonalcoholic fatty liver, commonly present in obesity [48]. Moreover, Eguchi et al. suggested that EVs from adipose tissue induce the recruitment and migration of macrophages, associated with obesity [57]. Furthermore, different EVs subpopulations (large and small), which differ in their lipid and protein content and that could be responsible for the inflammatory and metabolic alterations typical of obesity, have been found [54,66,67,68]. The mechanisms by which the content of EVs triggers any particular metabolic alteration, however, are not fully understood.

## 5. Extracellular Vesicles, Diet and Weight Loss

It is known that EVs are sensitive to many cellular stress situations, including sensitivity to nutrient deficiencies. Crewe et al. propose that EVs contain and transport proteins and lipids capable of modulating cellular signaling pathways between endothelial cells and adipocytes [69]. This transport event, which is made necessary in situations of fasting, refeeding, and obesity, is likely to be physiologically regulated so that EVs may participate in the tissue response to changes in the concentration of nutrients in the organism [69]. Thus, EVs could also be involved in the communication between adipose tissue and other cells. Gao et al. proposed that this involvement occurs as a means of communication between adipocytes and neurons [70]. This communication could modulate signaling pathways in the hypothalamus, which regulate appetite and weight gain.

A high-fat diet has been shown to cause morphometric changes (size) in EVs [56,71]. In addition, a high fat diet has been shown to cause changes in the expression of specific miRNAs contained in the EVs of hepatocytes, which, in turn, modulate the expression of various genes in other organs, such as the pancreas, causing hyperplasia in its islets [72]. Qi Fu et al. suggested that these changes caused by hepatocyte-derived EVs may be a compensatory measure of the B cells of the pancreatic islets under conditions of obesity and insulin resistance [72]. The sensitivity of EVs to insulin has also been shown by Eichner et al., who reported that, in humans, after receiving a glucose load, their levels of circulating EVs in the plasma decrease, and that this reduction could be associated with arterial stiffness, physical exercise, and insulin sensitivity [73]. Taking this research as a whole, it suggests that EVs are sensitive to specific modification of lipids and glucose in the diet. It is conceivable that, in humans without metabolic complications, EVs are also sensitive to the modifications of macronutrients, but research is needed to prove this.

Diet modification in obese subjects is a therapeutic tool for weight loss and improvements in metabolic biomarkers [74,75]. The evidence suggests that changes in the levels of EVs in the event of weight loss depend on the amount of weight lost and the degree of excess weight [62]. In adults with a BMI >35 kg/m2, weight loss of >25% does not show significant differences in the amount of EVs before and after weight loss [76]. In contrast, when compared to a control group, adults with an average BMI of 26 kg/m^2^ that presented weight losses of 5% had a significant decrease in the level of plasma EVs [44,75]. In addition, comparisons made between an excess vs. a normal weight group have shown a distinct EVs composition in the case of excess weight; in particular, differences in the profiles of proteins and nucleic acids (mainly miRNAS) involved in the development of cardiovascular diseases and diabetes [77,78,79].

The composition of EVs is known to be affected by changes in body weight, diet, and after bariatric surgery. Thrush reviewed the little information available on the changes in the characteristics of EVs that occur in successful- and unsuccessfully-treated subjects [80]. He reported that the EVs of obese patients that successfully respond to dietary treatment and weight loss stimulate oxidative metabolism in muscle cells to a greater extent than the EVs of obese patients resistant to treatment and weight loss [80]. This could help explain the variability that exists in response to dietary treatment and weight loss in obese subjects, but more evidence is needed to evaluate potential therapeutic goals to promote an appropriate dietary strategy for unsuccessfully treated patients.

Research has shown that the concentration and composition of EVs could change after bariatric surgery [47,81,82]. The physiological changes that occur after bariatric surgery also seems to bring about the modification of specific markers contained in EVs, which are implicated in the development of some alterations in adipose tissue, such as the free fatty acid transporter protein (FABP4) [82]. Given that FABP4 is mainly expressed in adipocytes, its changes in EVs after this surgical procedure could reflect the changes that occur in the adipose tissue, such as a reduction of adipocytes and of the total fat mass [82].

Although it is known that EVs are particularly sensitive to the early development of metabolic and cardiovascular disorders, the physiological role that they play in the pathogenesis of these disorders needs to be better understood so that they can eventually be used as biomarkers of these conditions, even before the first symptoms appear. Furthermore, more research is needed to validate their use in a clinical setting.

## 6. Extracellular Vesicles in Undernutrition

To date, the evidence pertaining to EVs in situations of undernutrition is limited and most of it has been developed in murine models or in vitro studies. For this reason, the available evidence on EVs and muscle depletion will be considered as a means to understand their possible involvement in undernutrition.

The skeletal muscle is one of the largest organ systems in the human body, representing ~40% of the body weight of an average adult. It plays a major role in maintaining homeostasis [4,83]. For instance, muscle mass plays an important role in the storage of glucose and amino acids, which are used by the body in stress or fasting situations, providing the backbone for the liver and gluconeogenesis process. Likewise, it plays an important role in the development of insulin resistance and other metabolic diseases [84]. However, there are situations that can affect the muscle mass composition and size, such as physical activity, chronic inflammation, sarcopenia, and malnutrition [83,85,86].

The communication between the muscle and other tissues such as adipose tissue, liver, and pancreas, is carried out through the release of myokines, “cytokines or peptides which are secreted by skeletal muscle cells and subsequently released into the circulation to exert endocrine or paracrine effects in other cells, tissues or organs” [87]. Researchers studying the transport mechanisms used by myokines to reach the bloodstream have suggested that they are transported by EVs [88]. This makes sense, as it is known that the muscle, like any other tissue in the body, releases EVs in the course of pathological conditions such as cancer [89], HIV [90], heart attacks [91], and kidney disease [92], or due to specific stimulus such as exercise [93]. Some authors have even suggested that the beneficial health effects that occur as a result of exercising are due to the content of myokines and miRNAs, among others, which are produced by muscle cells and transported via EVs [94,95]. Others have suggested that most of the circulating EVs during exercise are released by the muscle tissue, as it is the organ with the highest secretory activity [93,96,97]. The mechanisms for this process have not yet been fully understood. However, we could assume that the secretory activity of the muscle is not only reflected in the response to a specific physical activity, but also in response to another type of stimuli or condition that directly affects the proper functioning of muscle mass.

Based on the evidence that the muscle is responsible for the release of EVs during exercise, we could assume that this same process is replicated in other circumstances where the muscle is affected—among them, muscle loss in situations of undernutrition. Muscle mass is lost in undernourished adults in response to a deficient consumption of protein and to inflammatory response caused by pathological processes, which could manifest chronically or acutely [98]. The degree of inflammation is a key factor in the development and severity of undernutrition, including the development of extreme undernutrition (cachexia) [99]. Cachexia typically occurs in response to affections such as cancer, infectious diseases, or some autoimmune disorders [100]. Consequently, it leads to a greater propensity to develop infections, to a diminished response to pharmacological treatments, and to higher mortality [101].

The evidence on EVs and muscle wasting shows that these play a role in the development of cancer cachexia [102,103]. It has been proposed that the communication between cancer cells and other organs and tissues can cause an endocrine effect on muscle tissue. A mechanism has also been proposed in which EVs participate in muscle wasting, whereby the EVs content of HSP70/90 heat shock proteins activate, at the membrane level, the signaling pathway by TLR-4, which, in turn, triggers the degradation of regulatory and myofibrillar proteins [103]. It has also been hypothesized that the miRNA content in EVs promote myoblast death in murine cancer models via the TLR7 pathway [102]. Furthermore, various miRNAs in EVs, which are probably involved in the development of cachexia, have been identified and shown to participate in altering the signaling pathways that induce muscular apoptosis or dystrophy of this tissue [104,105], although the circulating levels of EVs with miRNAs are mainly attributed to the proliferation and communication of cancer cells and not specifically to muscle wasting. The identification of changes in EVs (content and composition) in neoplastic cachexia could potentially be a marker of muscle loss and wasting. Given this sensitivity of the EVs to this condition, which results in muscle loss and wasting, it could be hypothesized that undernutrition, free of cancer, could also bring about changes in EVs, which in turn could cause the muscle waste that typically accompanies undernutrition. To date, little is known about the changes in the size, characteristics, and content of EVs in undernutrition status when free of any disease.

Given the evidence that muscle is one of the most active tissues in releasing EVs into the blood stream during exercise and that inflammatory processes (e.g., cancer) affect the content of EVs, one could assume that the typical malnutrition abnormalities occur in response to the transport of proteins and of different nucleic acids transported by EVs. Thus, EVs could be responsible for or initiators of the typical muscle depletion observed in conditions of undernutrition. However, to our knowledge, there is no study that has tested this theory. Although much remains unknown about the physiological mechanisms that EVs follow in situations of undernutrition, if our theory is correct, the content and characteristics of EVs could serve as an early risk marker of muscle depletion in situations without associated comorbidities; for example, this could apply to patients with anorexia nervosa, as well as those undergoing muscle mass loss due to natural physiological changes associated with ageing, or due to lack of protein consumption for those with food insecurity. If EVs could be used as biomarkers of early risk of undernutrition, this would also contribute to the development of new therapeutic strategies for the prompt treatment of muscle depletion, which is a feature of this nutritional status. This is important, as current biomarkers only detect the condition in advanced stages.

## 7. Conclusions

The sensitivity of EVs to the cellular microenvironment could reflect early cellular alterations related to conditions of malnutrition (undernutrition and obesity). Despite the limited research to date on EVs in the area of nutrition, research in this area is increasing and could herald the discovery of mechanisms involved in the development of malnutrition and its pathological complications. This may lead to a better understanding of how EVs participate in or promote cellular signaling in malnutrition situations, which could help in the development of new strategies to treat them and their comorbidities. Thus, EVs could come to be excellent future biomarkers of early conditions associated with malnutrition and help to close the gap between the deterioration of health and time to detection. Of course, the use of EVs as biomarkers would not substitute the use of current ones. Rather, their use seeks a greater understanding of the physiological changes that occur prior to the development of health complications associated with malnutrition. This could also lead to them being used as a routine diagnostic tool in the future.

## Figures and Tables

**Table 1 nutrients-14-01002-t001:** Studies of extracellular vesicles in obesity.

Author, Year (Refs.)	Source of Isolation	EVs Size /Method of Isolation	EVs Classification	Specific Cell Marker	EVs Characteristics	Main Finding
Goichot, 2006 [42]	Plasma	NR ^A^	MP	Annexin V	Increase in EVs concentration (ug/mL)	Negative association with BMI
Esposito, 2006 [43]	Plasma	NR ^A^	MP	CD31, CD42	Increase in the number of EVs	Association with waist-hip ratio; C-reactive protein; HOMA-IR
Murakami, 2007 [44]	Plasma	NR ^A^	MP	CD41	Increase in the number of EVs	Association with BMI; waist circumference; subcutaneous body fat
Stepanian, 2013 [45]	Plasma	NR ^A^	MP	CD41, CD31, Annexin V	Increase in the number of EVs	The characteristics of EVs are independent of the metabolic syndrome
Kranendonk, 2014 [46]	Explant subcutaneous and omental adipose tissue	NR ^B^	EVs	CD9 Adiponectin	Association between the amount of EVs and WC and liver enzymes	Adipose tissue EVs can stimulate or inhibit insulin signaling at the liver level, depending on their adipokine content
Campello, 2015 [47]	Plasma	NR ^A^	MP	Annexin V, CD62, CD61, CD45	Increase in the number of EVs	Association with BMI, waist, fibrinogen, IL6, and FVIII; overproduction of EVs could induce the generation of thrombin
Koeck, 2015 [48]	Subcutaneous and visceral adipose tissue	50–100 nm ^C^	EXO	CD63	Increase in EVs concentration (ug/mL)	Higher BMI decreases the concentration of EVs
Togliatto, 2016 [41]	Visceral adipocyte stem cells primary culture	<1000 nm ^D^	EVs	CD63, CD81	No apparent change in size or quantity	Obesity impacts on the proangiogenic potential of EVs
Eguchi, 2016 [49]	Adipose tissue	NR^D^	EXO & ET	Perilipin A	Increase in EVs quantity	Association with biomarkers: glucose, insulin, and HOMA-IR; presence of perilipin A in adipocyte EVs
Mleczko, 2018 [50]	Plasma and adipocytes culture	100–150 ^D^	EXO	CD81, MHCI TSG101	No apparent change in size or quantity	EVs of obese subjects decrease insulin-stimulated 2-deoxyglucose caption in adipocytes
Mendivil, 2019 [51]	Plasma	<100 nm ^C^	EXO	ALIX	Increase in size of EVs	Association with BMI, TG, and % body fat
Santamarina, 2019 [52]	Plasma	<116 nm ^D^	EVs	NR	Smaller EVs size	Glucose, HOMA-IR, BMI, TG, HDL, and HA1c
Reza, 2020 [53]	Plasma	161 nm ^D^	EXO	CD63	No changes between groups were find	Participation in the insulin signaling pathway; increase in the intracellular content of TG and decrease the secretion of FGF21 in hepatocytes

BMI: body mass index; EVs: extracellular vesicles; EXO: exosomes; ET: ectosomes; method of isolation: ^A^ none reported, ^B^ sucrose gradient and ultracentrifugation, ^C^ synthetic polymer precipitation, ^D^ ultracentrifugation; MP: microparticles; MV: microvesicles; TG: triglycerides; WC: waist circumference; HOMA-IR: insulin resistance index; HDL: high density lipoprotein; HA1c: hemoglobin A1c.

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
