# Peer review of "Malnutrition and Biomarkers: A Journey through Extracellular Vesicles"

_nutrients, 2022, doi:10.3390/nu14051002_

Round 1

Reviewer 1 Report

The authors present the current research on the effect of malnutrition  (without associated diseases) on EVs and the likely role of EVs in the development of comorbidities associated with malnutrition.

There are a lot of papers on EVs in obesity, however I have not come across the combination of EVs and malnutrition in the literature, so I consider the topic to be original. I find the fragment about EVs and undernutrition especially interesting.

The paper creates a logical whole, contains correct citations. However I think, the language can be improved, even though I am not native.

My main remarks/questions. Yellow fragments come from the paper.

1. I would recommend to come across the paper once again and to remove some fragments that refer to the lack of data on the topic and the need for further research - I have the impression that this information is repeated too many times.

2. I would recommend adding arguments against the fact that EVs can be an early marker of malnutrition. EVs change in the state of fasting or after consumption of the meal… Are they too sensitive to many factors?

https://dx.doi.org/10.1093%2Fcdn%2Fnzab055_009

http://dx.doi.org/10.1055/s-0038-1642021

  1. The fragment about EVs as biomarkers takes up almost half of the I think, that the statements that EVs can be excellent biomarkers and can be used as routine diagnostic tool are too strong for now. Are these statements really conclusions from your work? I recommend to correct this section.
  2. Page 1, line 38-40

The term malnutrition encompasses disorders associated with deficit or excess in the consumption of nutrients, which are made manifest in conditions such as undernutrition and excess weight (obesity), known as the double burden of malnutrition.

“…which are made manifest in…”  has to be changed to “… which manifests in…”

  1. Page 3, Line 136

However, within the main mechanisms involved, the endosomal transport classification complex (ESCRT: endosomal sorting complexes required for transport)… - I find the word “classification” not correct here.

6. With this in mind, the aim of this review is to examine the current research on the effect of malnutrition (without associated diseases) on EVs and the likely role of EVs in the development of comorbidities associated with malnutrition and obesity.    The content in the brackets (page 2, line 89) is a repetition of the previous information in brackets (page 2, line 84-85) – the second information in brackets can be neglected, as the first information is very clear. The word “malnutrition” (page 2, line 90) should be changed to ‘undernutrition’.   

7. Page 3, line 99-100 bearing in mind that little is known about the influence of nutritional status on the sensitivity of EVs. “Sensitivity” is not a proper word here   

8. Page 4, 195-198 In fact, it is they who present a higher cardiovascular risk, since alterations in communication at the cellular level continue to occur, even without the apparent changes in parameters; traditional metabolic biomarkers, that is.                       - This sentence is not clear and needs to be changed    

9. Page 6, Table 1, last paragraph should be “No changes”    

10. Page 7, line 234-236

Santamaria et al. reports smaller sizes of EVs in obese women than in those with normal weight [52]. However, the number of small EVs isolated from obese and lean participants was found to be equivalent in obese and normal weight women [52].  It is not clear what explains these results, but Santamaria has suggested that the significant differences in glucose parameters and increased fatness in obese women, were responsible for the plasma EVs changes.

Not the most relevant findings of the study of Santamaria are mentioned here - size and amount are not as important as the composition.

  1. Page 7, line 258-259

Further, it has been shown that the activation of NF-KB also occurs in EVs of macrophages [65] –

NF-KB does not occur in EVs

  1. Page 8, 283-285

This transport event, which is made necessary in situations of fasting, refeeding and obesity, is likely to be physiologically regulated so that EVs may participate in the tissue response to changes in the state of systemic nutrients [69].

What does it mean “state of systemic nutrients”? Do you mean “general nutritional status”?

  1. Page 9, 362-368

However, we could assume that the secretory  activity of the muscle is not only reflected in the response to a specific physical activity, but also in response to other type of stimuli or condition that directly affects the proper functioning of muscle mass, such as undernutrition processes.

Based on the evidence that the muscle is responsible for the release of EVs during exercise, we could assume that this same process is replicated in other circumstances  where the muscle is affected; among them, muscle loss in situations of undernutrition.

I find the two sentences very similar.

  1. Page 9, 369-371

Muscle mass is lost in undernourished adults in response to a deficient consumption of protein and to inflammatory response caused by pathological processes, which could be made manifest chronically or acutely.

… which could be made manifest..  – not grammatically correct

  1. EVs could serve as an early risk marker of muscle depletion in situations without associated comorbidities What kind of patients with undernutrition do you mean? Do you think anorexia nervosa could be the clinical example of undernutrition without associated diseases?

Author Response

The authors present the current research on the effect of malnutrition (without associated diseases) on EVs and the likely role of EVs in the development of comorbidities associated with malnutrition.

There are a lot of papers on EVs in obesity, however I have not come across the combination of EVs and malnutrition in the literature, so I consider the topic to be original. I find the fragment about EVs and undernutrition especially interesting.

The paper creates a logical whole, contains correct citations. However I think, the language can be improved, even though I am not native.

My main remarks/questions. Yellow fragments come from the paper.

  1. I would recommend to come across the paper once again and to remove some fragments that refer to the lack of data on the topic and the need for further research - I have the impression that this information is repeated too many times.

ANSWER:

We appreciate your comment. However, we consider that it is important to signal at key places throughout the paper where further research is needed. We regard the identification of research gaps for the reader as integral to the revision.

2. I would recommend adding arguments against the fact that EVs can be an early marker of malnutrition. EVs change in the state of fasting or after consumption of the meal… Are they too sensitive to many factors?

https://dx.doi.org/10.1093%2Fcdn%2Fnzab055_009

http://dx.doi.org/10.1055/s-0038-1642021.
ANSWER:

Yes, EVs are sensitive to many factors. In Section 5. Extracellular vesicles, diet and weight loss, we have already included a brief description of the influence of some macronutrients on the characteristics of EVs, including on their content. Please see page 8, line 292: A high-fat diet has been shown to cause morphometric changes (size) in EVs [56].

The reference you suggested for this section has been added.

3. The fragment about EVs as biomarkers takes up almost half of the I think, that the statements that EVs can be excellent biomarkers and can be used as routine diagnostic tool are too strong for now. Are these statements really conclusions from your work? I recommend to correct this section. 

ANSWER:

Throughout the paper we have been very careful to be clear that there are still a lot of research gaps in relation to the use of EVs as biomarkers. At no instance do we make the claim that EVs can presently be used as routine diagnostic tool. Rather, the intention has been to show that the use of EVs as biomarkers can be an interesting possibility for the future. However, in attention to your observation, we have now stated this explicitly in the conclusion:  Page 10, line 418: Thus, EVs could come to be excellent future biomarkers of early conditions associated with malnutrition and help to close the gap between the deterioration of health and time to detection.

4. Page 1, line 38-40. The term malnutrition encompasses disorders associated with deficit or excess in the consumption of nutrients, which are made manifest in conditions such as undernutrition and excess weight (obesity), known as the double burden of malnutrition.

“…which are made manifest in…”  has to be changed to “… which manifests in…”

ANSWER:

Thank you for the observation. We have fixed this as per your suggestion.

5. Page 3, Line 136. However, within the main mechanisms involved, the endosomal transport classification complex (ESCRT: endosomal sorting complexes required for transport)… - I find the word “classification” not correct here.

ANSWER:

Thank you. We have corrected this. It now reads: “…endosomal transport sorting complex...”

6. With this in mind, the aim of this review is to examine the current research on the effect of malnutrition (without associated diseases) on EVs and the likely role of EVs in the development of comorbidities associated with malnutrition and obesity.  The content in the brackets (page 2, line 89) is a repetition of the previous information in brackets (page 2, line 84-85) – the second information in brackets can be neglected, as the first information is very clear. The word “malnutrition” (page 2, line 90) should be changed to ‘undernutrition’.   

ANSWER:

Thank you. We have corrected this in the text (page 2, lines 87-90). It now reads: With this in mind, the aim of this review is to examine the current research on the effect of malnutrition on EVs and the likely role of EVs in the development of comorbidities associated with undernutrition and obesity.

7. Page 3, line 99-100 bearing in mind that little is known about the influence of nutritional status on the sensitivity of EVs.“Sensitivity” is not a proper word here   

ANSWER:

On page 2, line 76-80 we clarified the sense in which the word "sensitivity" was to be understood in terms of characteristics of extracellular vesicles: “Henceforth, when we speak of the sensitivity of EVs we are referring to this seemingly highly calibrated response, which is manifested with variations in size, quantity and content of EVs. Indeed, due to the response of EVs to situations of cellular stress, their use as potential biomarkers in some pathological conditions has been suggested, such as in neurodegenerative, cardiovascular, chronic-degenerative diseases and some types of cancer  [9–12].”  Thus, we believe that it’s use is appropriate here. 

8. Page 4, 195-198In fact, it is they who present a higher cardiovascular risk, since alterations in communication at the cellular level continue to occur, even without the apparent changes in parameters; traditional metabolic biomarkers, that is.                     - This sentence is not clear and needs to be changed    

ANSWER:

We have corrected this sentence to make it clearer. It now reads: Page 4, line 195-198. “In fact, it is the MHO who present a higher cardiovascular risk, since alterations in communication at the cellular level continue to occur, even without the apparent changes in traditional metabolic biomarkers [37–41].”

9. Page 6, Table 1, last paragraph should be “No changes”.

ANSWER:

Thank you. It has been corrected.

10. Page 7, line 234-236. Santamaria et al. reports smaller sizes of EVs in obese women than in those with normal weight [52]. However, the number of small EVs isolated from obese and lean participants was found to be equivalent in obese and normal weight women [52].  It is not clear what explains these results, but Santamaria has suggested that the significant differences in glucose parameters and increased fatness in obese women, were responsible for the plasma EVs changes.

Not the most relevant findings of the study of Santamaria are mentioned here - size and amount are not as important as the composition.

ANSWER:

We have added the information regarding the microRNA from the Santamaria paper. Thank you (Page 7. Line 234, 235).

11. Page 7. Line 258. Further, it has been shown that the activation of NF-KB also occurs in EVs of macrophages [65] – NF-KB does not occur in EVs

ANSWER:

You are absolutely correct in your observation. We changed the expression to: “Further, it has been shown that the NF-KB can also be stimulated by the EVs of macrophages” (Page 7. Line 259, 260).

12. Page 8, 283-285. This transport event, which is made necessary in situations of fasting, refeeding and obesity, is likely to be physiologically regulated so that EVs may participate in the tissue response to changes in the state of systemic nutrients [69].

What does it mean “state of systemic nutrients”? Do you mean “general nutritional status”?

ANSWER:

The term “state of systemic nutrients” was a term that came directly from the cited reference. However, we agree with the reviewer that it is not clear. Therefore, we have now replaced it with: “…changes in the concentration of nutrients in the organism” (page 8, line 287, 288).

13. Page 9, 362-368. However, we could assume that the secretory  activity of the muscle is not only reflected in the response to a specific physical activity, but also in response to other type of stimuli or condition that directly affects the proper functioning of muscle mass.

Based on the evidence that the muscle is responsible for the release of EVs during exercise, we could assume that this same process is replicated in other circumstances  where the muscle is affected; among them, muscle loss in situations of undernutrition. I find the two sentences very similar.

ANSWER:

The sentences are similar but not the same. The first sentence refers to the muscle in general, while the second sentence integrates the information but from the perspective of EVs. However, we have removed the mention of undernutrition from the first sentence to make it less repetitive (page 9, line 367).  

14. Page 9, 369-371. Muscle mass is lost in undernourished adults in response to a deficient consumption of protein and to inflammatory response caused by pathological processes, which could be made manifest chronically or acutely.

… which could be made manifest..  – not grammatically correct

ANSWER:

Thank you for the observation. We have changed it to: “…which could manifest chronically or acutely.”

15. …EVs could serve as an early risk marker of muscle depletion in situations without associated comorbidities What kind of patients with undernutrition do you mean? Do you think anorexia nervosa could be the clinical example of undernutrition without associated diseases?

ANSWER:

It is not just in subjects with anorexia nervose, but also in subjects that show a loss of muscle mass due to ageing or lack of protein consumption due to food insecurity. We have added some examples following your highlighted sentence (page 10, line 408-410) to make it clearer for the reader:

“For example, this could apply to patients with anorexia nervosa, as well as those undergoing muscle mass loss due to natural physiological changes associated with ageing, or due to lack of protein consumption for those with food insecurity.”

Reviewer 2 Report

This is an exciting review in which the authors have drawn up a novel topic. It is important to stress that the link between Malnutrition and EVs is, at least at this point- insufficiently reported in many scientific investigations. The manuscript is well structured, and sufficient information about the previous studies' findings is presented to follow the review topic. However, from my point of view, there are just a few drawbacks associated with the manuscript, such as the grammar of English (see lines 195-200 and 225-228). I also want to suggest that authors cite more relevant and recent literature regarding the efficacy of EVs compared with other systems by cutting out the first part about the definition and biogenesis of EVs that is well known.

Author Response

Comments and Suggestions for Authors.

This is an exciting review in which the authors have drawn up a novel topic. It is important to stress that the link between Malnutrition and EVs is, at least at this point- insufficiently reported in many scientific investigations. The manuscript is well structured, and sufficient information about the previous studies' findings is presented to follow the review topic. However, from my point of view, there are just a few drawbacks associated with the manuscript, such as the grammar of English (see lines 195-200 and 225-228). I also want to suggest that authors cite more relevant and recent literature regarding the efficacy of EVs compared with other systems by cutting out the first part about the definition and biogenesis of EVs that is well known.

 ANSWER.

Thank you for your comments. We have fixed the grammatical issues you have highlighted and has been revised by a native English speaker, who is also a competent researcher.  In relation to your suggestion to remove the first part about the definition and biogenesis of EVs, we consider that it is important to keep as it helps to contextualize the paper as a whole and may be of much use for people new to the field. In addition, it is important to continue promoting the use of the criteria proposed by the International Society of Extracellular Vesicles. In relation to the last point, at present, there is not sufficient validated information to allow us to compare the possible use of EVs with other systems and biomarkers.